# A Cross-Sectional Study on Disparities in Unmet Need among Refugees and Asylum Seekers in Thailand in 2019

**DOI:** 10.3390/ijerph18083901

**Published:** 2021-04-08

**Authors:** Watinee Kunpeuk, Sataporn Julchoo, Mathudara Phaiyarom, Pigunkaew Sinam, Nareerut Pudpong, Rapeepong Suphanchaimat

**Affiliations:** 1International Health Policy Program, Ministry of Public Health, Tiwanon Road, Nonthaburi 11000, Thailand; sataporn@ihpp.thaigov.net (S.J.); mathudara@ihpp.thaigov.net (M.P.); pigunkaew@ihpp.thaigov.net (P.S.); nareerut@ihpp.thaigov.net (N.P.); rapeepong@ihpp.thaigov.net (R.S.); 2Division of Epidemiology, Department of Disease Control, Ministry of Public Health, Nonthaburi 11000, Thailand

**Keywords:** unmet need, refugees, asylum seekers, health policy

## Abstract

The health of urban refugees and asylum seekers (URAS) in Thailand has been under-researched compared with other groups of non-Thai populations, especially on the topic of unmet need. The aim of this study is to examine the level of unmet need among URAS in Thailand, focusing on out-patient (OP) and in-patient (IP) care. A cross-sectional study was conducted between October and December 2019. Stratified random sampling was undertaken and a total of 181 participants were included. A bivariate analysis was used to explore the level of unmet need among different URAS groups. Multivariate logistic regression was undertaken to examine the association between potential correlations and unmet need of IP and OP care. The majority of participants were female and aged below 30 years, with a low educational background and poor economic status. Most of them had experienced an illness in the past month before the interview, and some suffered from chronic diseases. The prevalence of both OP and IP unmet needs was more frequent among URAS from Arab countries. Furthermore, being from Arab countries indicated a strong link with OP and IP unmet need. Additionally, having illness over the past six months and chronic diseases were found to be significant determinants of IP unmet need. Our analysis showed that nationality had a strong association with both IP and OP unmet need, especially among those from Arab countries. Therefore, culturally appropriate health services should be considered to promote healthcare access among diverse groups of URAS. Further qualitative studies on barriers to accessing OP and IP care, such as communication, interpretation, and social dimensions, are required.

## 1. Introduction

Access to healthcare is a leading indicator of how well health systems perform in different contexts [1,2]. While complexity is embedded in the core concept of access to care, health service utilization and unmet need have been used as the best proxies to measure it [3,4]. Given the migration crisis, vulnerable groups such as refugees and asylum seekers (URAS) are facing great barriers in accessing healthcare. Both physical and mental health problems—arising from the countries of origin, during the transition to, and in the period of arrival in host countries—are significant among them [5]. Various factors can exacerbate their health status, including a lack of inclusive policies, financial hardship, illegal status, and linguistic and cultural barriers [6,7]. Many countries worldwide are unable to guarantee medical and social services to URAS due to legal restrictions and bureaucracy that limit their rights for health and social welfare [4].

URAS tend to have a higher need for healthcare compared to other groups. Mental health disorders, including depression and post-traumatic stress disorder (PTSD), are frequent among them [2]. A study in the Unites States (US) showed that the prevalence of anxiety, depression, and emotional stress among Iraqi refugee women was approximately 50%, and 31% experienced PTSD [8]. Health screening in Australia revealed that 50% of asylum seekers faced mental illness, with about 25% screening positive for PTSD [9]. Turkey was also found to host a high influx of refugees, with the highest figure for Syrians, Iraqi, and Afghanistan [10]. Evidence suggested that infectious diseases were pronounced among Syrian refugees in Turkey, and preventive services were inadequate in the refugee camps [11]. Furthermore, a survey among newly arrived adult immigrants residing in the US found that about 60% of the participants had at least one chronic condition, such as high cholesterol, hypertension, overweight/obesity, and diabetes [8]. A study using a health access and utilization survey (HAUS) conducted by the United Nations High Commissioner for Refugees (UNHCR) in 2015 showed that 26.7% of URAS in Malaysia had chronic conditions, and that they were unable to find treatment [12]. Moreover, although the majority of URAS pregnant women had made at least one antenatal visit, almost half faced difficulties in accessing antenatal care. This study also found that 45.2% of healthcare visits were at government facilities, 43.3% were at private health facilities, and 11.5% were at non-governmental organization (NGO) facilities [7,12]. Regarding chronic diseases, the HAUS 2018, conducted among 400 Syrian refugee households in Jordan, suggested that 27% had hypertension, 19% had diabetes, and 14% had asthma or chronic obstructive pulmonary disease [13]. In terms of chronic disease treatments, 74% of them found that they could not access medicines, and of this number, 52% found that they could not afford the medicine cost, and about 19% faced the unavailability of essential medicines [13]. However, the percentage of Syrian refugees with chronic diseases who were unable to access medical services decreased from 39% in 2017 to 22% in 2018; the main reason was affordability of medical costs (49%) [13].

The estimated number of displaced individuals across the globe in 2019 was 79.5 million, and about half were cross-border people seeking refuge and asylum [14]. Over the past decade, the number of URAS in the Asia and Pacific region has increased by 3%. Most URAS in the region are stateless refugees who traveled from Myanmar to Bangladesh in 2017. There have also been a number of Afghan refugees in Pakistan and the Islamic Republic of Iran in the region for over 40 years [14]. The United Nations High Commissioner for Refugees (UNHCR) reported that there were 573,518 persons of concern in Thailand by the end of 2019, with 93,333 Myanmar refugees in camps, 5070 urban refugees and asylum seekers, 474,888 stateless people, and 227 others [13]. As Thailand is not a signatory of the 1951 Convention Relating to the Status of Refugees [15], URAS are recognized as marginalized people and are invisible in the society. URAS are also facing risks of arrest, detention and deportation, and limited opportunities for work and education [16]. However, the Thai government has made progress in enhancing their rights. For example, in January 2019, the Royal Thai Government committed to end the detention of refugees’ and asylum seekers’ children and issue an approval for screening mechanisms to identify people who need international protection from economic migrants [13].

Generally, URAS often experience difficulties with the inclusiveness of healthcare services. The increasing number of URAS puts pressure on the healthcare system; essentially, the greater the level of URAS diversity, the wider the health inequality gap in host nations [17]. Given that they are unauthorized entrants and are not legally recognized, URAS in Thailand are not covered by mainstream public health insurance. Humanitarian assistance, including healthcare services for URAS, is mainly provided by the UNHCR and NGOs. However, due to a limited budget coupled with the continuous new arrival of URAS, the UNHCR revised medical guidelines on healthcare services in 2014 to have stricter limitations for URAS with health concerns [18]. The revision removed all full cash reimbursements for OP and IP care. Instead, medical assistance was prioritized and currently covers only certain chronic diseases, life-saving interventions, and emergencies. The Refugee Centre Medical Clinic was made available to provide counseling to people seeking medical treatment at affordable public health facilities. Furthermore, medical referrals to public health facilities need to be provided for emergency or life-threatening cases only [18].

There is limited migrant health literature that explores how URAS access and use healthcare in Thailand. This research appears to be one of the first studies aiming to quantify the degree of unmet need among different nationality groups of URAS in Thailand, with a focus on OP and IP care. This evidence can be used to fill the gap in a systematic evaluation on healthcare access among URAS, especially in the context of low- and middle-income countries such as Thailand.

## 2. Materials and Methods

### 2.1. Study Design

This study was part of the 2019 cross-sectional survey among URAS in Thailand conducted by International Health Policy Program (IHPP), Thailand [19], aiming to explore the degree of healthcare access among URAS in comparison with the Thai population. The study was performed between October and December 2019. A sampling process was undertaken by using the URAS roster with support from the Bangkok Refugee Centre (BRC), a partner of UNHCR providing assistance on education, health, and vocational training for URAS [20]. Only the top-ten most common nationalities in Thailand were the focus of this study. These included the Pakistani, Vietnamese, Cambodian, Somali, Afghan, Palestinian, Chinese, Sri Lankan, Iraqi, and Syrian. The total number of URAS listed in the roster was 3021, and 206 samples were drawn to participate in the study. Of the 206 samples, 181 completed the survey.

### 2.2. Sample Size Calculation and Data Collection

The sample size was calculated by using the prevalence of unmet need for healthcare among URAS and the formula was shown below. In this study, we replaced *P* with unmet need prevalence in URAS reported by Busetta et al. [21] at 11% (or 0.11 in the formula). Although Busetta et al.’s research was conducted in Italy, the prevalence of unmet need could be applied in this research because there was no information on unmet need prevalence of URAS in Thailand. Moreover, as the public sector is the main provider for health systems in both Thailand and Italy, unmet need prevalence regarding the study by Busetta et al. [21] was assumed to be applicable in this context. After applying all variables in the formula, the final number derived from calculation amounted to 180 samples, considering a 20% non-response rate and incomplete information.
$$n=\;\frac{Z^{2}P\left(1\;-\;P\right)}{d^{2}}$$
where *n* = Sample size

*Z* = *Z* statistics for the level of confidence

*P* = Expected prevalence or proportion

*d* = Precision (in proportion of one; if 5%, *d* = 0.05).

The sample collection was conducted by using stratified random sampling with probability proportional to size and took into account age group, sex, and nationality. The BRC undertook the participant recruitment themselves; the number of participants reached 206, which was higher than the calculated sample size above. As some participants refused to participate or complete the survey (23 did not participate and two did not finish the questionnaire), the final number of participants who completed the questionnaire was reported at 181. For the data collection process, participants were invited to the BRC to complete the survey, and a financial incentive was provided for travelling costs (US $9 per person). Some participants had a phone interview due to transport hardship. For participants aged below 15 years, their parents or legal guardians undertook the questionnaire on their behalf. When some interviewees faced a reading barrier, a verbal interview was applied to facilitate their response. The questionnaire was also translated into different URAS languages to accommodate respondents and increase the number of valid responses. For each URAS group, a specific coordinator, who was a volunteer of the BRC members, contacted URAS and distributed the questionnaires. There was also a preparation meeting for the interview with engagement from coordinators and research team members. Each respondent spent about 30 min for the whole questionnaire.

### 2.3. Questionnaire Design

The questionnaire was designed based on relevant questions from the Thai Health Welfare Survey (HWS), which consisted of two parts: (i) demographic characteristics of participants and (ii) unmet need for healthcare. Consultative meetings were undertaken over two rounds to ensure content validity with engagement from research team members, academic researchers, and BRC members. Some variables were re-categorized to allow for more in-depth data analysis. Age was categorized into three groups (≤15 years, >15 but ≤60 years, and >60 years). Household income was divided using the monthly household income cut-off in Bangkok of THB 45,707 (US $1428) [19,22]. Unmet need of healthcare services was based on self-assessment for OP and IP care with the same question internationally applied for unmet need assessment [23], “During the last 12 months, he or she had felt unwell and needed healthcare but did not receive it.” If the participant experienced unmet need, they were further asked about the reason for it. Finally, URAS category was rearranged into three groups: (i) Southeast Asia (ASEAN), including Cambodia and Vietnam; (ii) Arab countries (ARAB), including Iraq, Palestine, Somalia, and Syria; and (iii) Others (OTHERS), including Afghanistan, China, Sri Lanka, and Pakistan.

### 2.4. Data Analysis

For descriptive statistics, categorical data were reported in absolute and relative frequencies and continuous variables in median and interquartile range (IQR) as the data were found to have a right-skewed distribution. Inferential statistics were also conducted to explore differences between each demographic variable among various groups of URAS, and to find associations between unmet need for healthcare and each variable of interest. Bivariate analysis using chi-square or Fisher’s exact test was employed to examine the relationship between two categorical variables, and the Kruskal–Wallis test was applied for continuous variables. Logistic regression analyses were used to predict associations between OP and IP care (as dependent variables) and selected independent variables. Independent variables included in the multiple logistic regression model comprised nationality, sex, age, education level, economic level, being sick over the past month, and chronic diseases. Crude and adjusted odds ratios (OR) with 95% confidence interval (95% CI) were reported as the final outputs. A two-tailed *p*-value of 0.05 was considered statistically significant. All analyses were conducted on weighted data using STATA V.13.1 (StataCorp LLC., College Station, TX, USA) (see Appendix A).

### 2.5. Ethics

The study received ethical approval from the Institute for the Development of Human Research Protections in Thailand (IHRP 595/2562) prior to data collection. Written consent was obtained from all participants. For those uncomfortable with written consent or illiterate participants, verbal consent was used instead. All participants were well-informed about the project and were re-assured to feel free to withdraw from the study at any time.

## 3. Results

### 3.1. Demographic Characteristics

Of the 181 participants from the URAS roster, 62 participants were from ASEAN countries, 28 were from ARAB countries, and the remaining 91 were from the OTHERS countries. Females (52%) were in a slightly higher proportion than males (48%). About 40% of URAS had a primary education degree, and the ARAB URAS had the highest figure for tertiary education (24%). The median age of URAS from the ARAB group was 29 years, which was higher than that of the OTHERS (23 years) and ASEAN (18.5 years) groups. The median household income per month in the ARAB group was higher than that of the OTHERS and ASEAN at THB 7000 (US $233), THB 6000 (US $200), and THB 5000 (US $166), respectively. When using the monthly household income for Thais in the Bangkok area of THB 45,707 (US $1524) as the cut-off point, more than 80% of all URAS were below that figure. More than half of the participants responded that they had been ill over the past six months. Approximately one quarter of the OTHERS group experienced chronic diseases. The statistical test showed that there was no statistically significant difference for all variables between each group of URAS, except for being sick over the past months (*p* < 0.05) (see Table 1).

### 3.2. Unmet Need Prevalence

Statistical significance was found in unmet need for OP and IP care among the three URAS groups (*p* < 0.001) (Table 2). The ARAB group experienced the highest prevalence of OP unmet need at 86%, followed by the OTHERS group at 59%, and the ASEAN group at 32%. Likewise, the largest unmet need for IP care was found in the ARAB group at 62%, which was higher than the OTHERS group at 29%, and the ASEAN group at 13%, respectively.

### 3.3. Bivariate and Multivariable Logistic Regression Analyses

The results from the multivariable logistic regression showed that nationality was significantly associated with unmet need for OP care. Being in the ARAB group indicated a strong link with OP unmet need and was higher than the ASEAN group in terms of both unadjusted and adjusted odds (unadjusted OR = 12.6, 95% CI 3.85–41.21, *p* < 0.001; adjusted OR = 14.68, 95% CI 3.93–54.91, *p* < 0.001). Moreover, after adjusting for sex, age, economic level, being sick over the past six months, and having chronic diseases, being in the OTHERS group demonstrated a significant association with OP unmet need (OR = 2.69, 95% CI 1.26–5.77, *p* < 0.05). Furthermore, females were more likely to experience unmet need than males but the association was not significant after including sex in the multivariable logistic regression. Elderly people tended to have higher odds of OP unmet need than people aged below 15 years, although the association between age and OP unmet need did not show any statistical association. URAS for whom the income level was below the cut-off of THB 45,707 experienced greater OP unmet need (adjusted OR = 3.5, 95% CI 1.04–11.68, *p* < 0.05). Furthermore, the odds of OP unmet need were higher among URAS who were ill within the six months prior to the interview, with high statistical significance (adjusted OR = 4.32 95% CI 2.02–9.24, *p* < 0.001). URAS with chronic diseases tended to have greater OP unmet need but there was no statistical significance after adjusting with other independent variables, see Table 3.

For IP unmet need, like OP, being in the ARAB group had a strong relationship with unmet need of IP care. The ARAB group was likely to have 12 times greater IP unmet need than the ASEAN group (adjusted OR = 12.93, 95% CI 3.92–42.65, *p* < 0.001). URAS from the OTHERS group tended to experience higher IP unmet need relative to the ASEAN group but the association was not strongly significant. The relationship between female URAS and OP unmet need had no statistical significance. This trend was also found in the relationships between age group and IP unmet need, and economic level and IP unmet need. However, having illness over the past six months and chronic diseases were strongly linked to IP unmet need. The analysis indicated that URAS experiencing illness before the interview seemed to face IP unmet need about three times higher than those without illness (adjusted OR = 3.49, 95% CI 1.34–9.08, *p* < 0.05). Furthermore, URAS with chronic diseases had greater odds of IP unmet need relative to those without chronic diseases by approximately three times (adjusted OR = 2.84, 95% CI 1.06–7.6, *p* < 0.05) (see Table 4).

## 4. Discussion

To our knowledge, this is the first study to examine unmet need of OP and IP care among URAS in Thailand. The results suggest that the majority of URAS traveled from countries including Afghanistan, China, Sri Lanka, and Pakistan (or the OTHERS), followed by the ASEAN and ARAB regions. Most URAS were female and aged below 30 years. The majority had a primary education background, were of low economic status, and reported that they were ill during the past month before the survey was conducted.

Unmet need of OP and IP care showed a similar trend overall, with the ARAB group facing the highest prevalence of unmet need, followed by the OTHERS and ASEAN groups. In addition, unmet need of OP care was higher than IP care in all URAS groups. It can be postulated that, for OP services, it depends on the operations of the BRC and healthcare support, which is considered on a case-by-case basis. Consequently, the majority of URAS are unable to receive OP services. IP care, however, is often prioritized and therefore it is likely that URAS with severe illness can access IP services. The situation of health service access among URAS is in stark contrast with the Thai populations. In Thailand, according to the National Health Security Office—the main governing body of the Universal Coverage Scheme for most Thai populations, the amount of OP and IP utilization in Thais has been increasing. In fiscal year 2018, there were 184.56 million visits and the utilization rate was 3.845 visits/person/year [24]. For IP services, the number of IP usage was 6.22 million with a rate of 0.127 admissions/person/year. The annual prevalence of unmet need for OP and IP services in 2010 was minimal, only 1.4% and 0.4%, respectively [25]. The most common reason for unmet need for healthcare in this population is likely to be associated with a lack of funds for treatment cost [8]. Several other country studies also showed barriers of healthcare access. For instance, a survey of healthcare access showed that Iraqi refugees who had resettled in the US faced delays in medical treatment within the last 12 months due to a lack of health insurance, lack of interpretation, and financial hardship—meaning that they could not afford treatment [8]. Qualitative evidence by Herrel et al. [26] suggested that Somali women who experienced healthcare services for pregnancy and childbirth in Minnesota faced challenges from healthcare staff including discrimination and having less sensitive care due to language barriers.

The results highlight that being in the ARAB group and being sick over the past month are strong determinants of experiencing unmet need in both OP and IP care. A scoping review conducted by Mangrio and Sjögren Forss [2] illustrated that some asylum seekers in Australia could not afford the medical consultation fee, making it difficult for them to see the doctor. Some received support from charity organizations, but access to healthcare services was limited and required long waiting times, particularly in the emergency room [2]. As a result, it had an impact on physical and mental health conditions, and in some cases this could lead to serious health illness [27]. Therefore, a special healthcare system may be required specifically for the ARAB group URAS. Taylor et al. [8] confirmed that Iraqi refugees in the US showed a high prevalence of chronic conditions although they had relatively high access to healthcare. Access to healthcare among Afghan refugees in Australia was also explored in a study by Omeri et al. [28], which showed that language problems and culturally sensitive issues were the main barriers preventing access to healthcare. This included a lack of familiarity with the Australian healthcare system; a lack of interpreters and discrimination caused by language barriers; stereotype challenges related to religion and Islamic attire; gender issues; travel costs; long waiting times; and poor health-related information [28]. As such, appropriate methods of health screening, prevention, and treatment by embracing cultural differences to improve access to healthcare and follow-up, especially among the ARAB URAS in Thailand, should be considered.

The design of culturally appropriate healthcare systems is also crucial. Some studies have suggested strategies for providing appropriate healthcare by taking into account cultural issues [2,28,29]. Culturally appropriate care is considered important and needed for dealing with challenges, particularly for symptomologies, diagnoses, and medical terminology [29]. For chronic diseases, culturally sensitive approaches can provide a better understanding to develop health interventions for health promotion and lifestyle changes [28]. To strengthen cultural awareness and to understand and overcome cultural barriers, collective efforts are needed, especially among community members [30]. Integration of cultural safety principles into healthcare practices and research may also help embrace social and cultural diversity of URAS in the Thai health system [29].

The results from multivariate logistic regression showed that being in the ARAB group and the OTHERS group, poor economic status, and having illness over the past month had a strong relationship with IP care. Moreover, being in the ARAB group, being sick over the past month, and having chronic diseases were significantly associated with OP unmet need. However, this study’s results differed from the previous research by Wu et al. [31], which reported that age, gender, and education are important factors leading to the unmet need of immigrants in Canada. Our study showed that the relationship between sex and unmet need is not statistically significant. This might be because both male and female URAS may have low opportunities for accessing healthcare in Thailand indifferently. To identify more specific needs of URAS, comparative studies in relation to gender-sensitive issues should be explored

The findings of this study provide novel insights of unmet need for healthcare among URAS in Thailand by using stratified random sampling at the individual level. Nevertheless, some limitations need to be considered. Firstly, there was a lack of opportunities to interview refugees and asylum seekers directly as some questionnaires were answered by surrogates. Therefore, reliance on unmet need assessment from some respondents could be challenging. Secondly, the calculation of samples used a parameter cited from the Italian study [21]. This is because it used the same unmet need questionnaire with this study and the studies in the topic of unmet need for URAS in Asia are lacking. The authors were aware that this is not the best approach for sample size calculation. However, it is the most practical approach that this study could exercise with limited resources and timeframe. This point also reflects a dire need for research in URAS in Asia. Moreover, the readers should be reminded about the difference between the healthcare system of Italy and Thailand. In Italy, the main healthcare system is known as Servizio Sanitario Nazionale (SSN), which is a regionally based health service [32]. It provides free-of-charge universal coverage at the point of care. Most local health facilities are governed by the regional governments, while the national level ensures that the general objectives and principles of the national healthcare system are met. In Thailand, the majority of health facilities are affiliated to the central authority—namely, the Ministry of Public Health. However, the systems in both countries share some similar features. For instance, both systems are mainly financed by general tax, and the insurance benefits for the nationals in both countries are quite comprehensive—ranging from basic treatment and health promotion to high-cost care. Thirdly, memory bias could have occurred during the interview regarding unmet need as respondents were asked to provide their healthcare access history over the past 12 months. Lastly, there was no physical visit during the interview due to an ethical concern about the residential confidentiality or URAS. As such, these limitations can lead to a gap in evidence on household infrastructure and conditions. Moreover, no formal clinical diagnosis data were collected in this study. Future studies that compare the need of services of URAS with the official diagnosis by doctors are of great value. However, this idea needs much more time and resources, and should be exercised by a concerted effort between the Thai Ministry of Public Health, BRC, and UNHCR—not just by the independent research team, as in this study.

## 5. Conclusions

URAS in Thailand are considered marginalized and vulnerable in relation to healthcare access. Without any health insurance coverage, the majority of URAS experience unmet need for healthcare in terms of both IP and OP care. Apart from a common financial barrier leading to unmet need, the nationality of URAS is a strong determinant. In this study, the ARAB group of URAS tended to experience a greater level of unmet need in comparison with the ASEAN and OTHERS groups. Due to a variety in the degree of unmet need, culturally appropriate healthcare should be taken into account to ensure that all diverse groups of URAS can access healthcare in Thailand. Given that URAS are likely to have greater risk from mental and physical illness, the healthcare system in host countries may need to be redesigned, especially for responses to chronic diseases. Further qualitative studies are needed to explore communication barriers and other social challenges that diverse groups of URAS face when trying to access healthcare.

## Abbreviations

URASUrban refugees and asylum seekersIPIn-patientOPOut-patientPTSDDepression and posttraumatic stress disorderUSthe Unites States HAUSHealth access and utilization surveyUNHCRUnited Nations High Commissioner for RefugeesIHPPInternational Health Policy Program, ThailandNGOsNon-governmental organizationsBRCBangkok Refugee CentreASEANSoutheast Asia including Cambodia and Vietnam ARABArab countries including Iraq, Palestine, Somalia, and SyriaOTHERSCountries including Afghanistan, China, Sri Lanka, and PakistanIQRInterquartile rangeOROdds ratio

## Figures and Tables

**Table 1 ijerph-18-03901-t001:** Demographic characteristics of participants.

Demographic Characteristics of Participants (*n* = 181)	ASEAN ^1^ (*n* = 62)	ARAB ^2^ (*n* = 28)	OTHERS ^3^ (*n* = 91)	*p*-Value	Statistics
Sex				0.479	Chi-square
Female	31 (50)	17 (62.96)	46 (50.55)		
Male	31 (50)	10 (37.04)	45 (49.45)		
Education—*n* (%)				0.433	Chi-square
Up to primary	17 (47.22)	7 (41.18)	33 (47.14)		
Up to secondary	17 (47.22)	6 (35.29)	29 (41.43)		
Degree or above	2 (5.56)	4 (23.53)	8 (11.43)		
Median age—year (P25, P75)	18.5 (9.0, 38.7)	29 (12.5, 42.3)	23 (11.1, 37.1)	0.711	Kruskal Wallis Test
Mean age (SD)	23.9 (16.6)	26.6 (16.9)	25.0 (17.5)		
Age group—*n* (%)				0.125	Chi-square
<15 years	26 (41.94)	10 (33.33)	39 (42.86)		
>15 and <60 years	34 (54.84)	16 (53.33)	50 (54.95)		
>60 years	2 (3.23)	4 (13.33)	2 (2.20)		
Median household income—Baht (IQR) ^4^	5000 (4000)	7000 (4000)	6000 (5500)	0.156	Kruskal Wallis Test
Household economy (THB 45,707 (US $1524) as the cut-off point for household income)				0.916	Fisher’s exact test
Above average	9 (14.52)	4 (13.33)	11 (12.09)		
Below average	53 (85.48)	26 (86.67)	80 (87.91)		
Being sick over the past month				0.017	Fisher’s exact test
No	30 (48.39)	8 (28.57)	24 (26.37)		
Yes	32 (51.61)	20 (71.43)	67 (73.63)		
Chronic diseases				0.227	Fisher’s exact test
No	50 (80.65)	25 (89.29)	66 (74.16)		
Yes	12 (19.35)	3 (10.71)	23 (25.84)		

^1^ ASEAN: Southeast Asia countries including Cambodia and Vietnam; ^2^ ARAB; Arab countries including Iraq, Palestine, Somalia, and Syria; ^3^ Others: other countries including Afghanistan, China, Sri Lanka, and Pakistan; ^4^ IQR: interquartile range.

**Table 2 ijerph-18-03901-t002:** Unmet need prevalence of Out-patient (OP) and In-patient (IP) care.

Country	Unmet Need (*n* (%))		Total
	No Unmet Need	Having Unmet Need	
OP care			
ASEAN	42 (67.74)	20 (32.26)	62 (100)
ARAB	4 (14.29)	24 (85.71)	28 (100)
OTHERS	37 (40.66)	54 (59.34)	91 (100)
IP care			
ASEAN	54 (87.10)	8 (12.90)	62 (100)
ARAB	10 (38.46)	16 (61.54)	26 (100)
OTHERS	126 (71.26)	25 (28.74)	87 (100)

ASEAN: Southeast Asia countries including Cambodia and Vietnam; ARAB; Arab countries including Iraq, Palestine, Somalia, and Syria; Others: other countries including Afghanistan, China, Sri Lanka, and Pakistan.

**Table 3 ijerph-18-03901-t003:** Bivariate analysis and multivariate logistic regression for unmet need of out-patient (OP) care.

Factors	Bivariate Analysis		Multivariable Logistic Regression	
	Crude OR (95% CI)	*p*-Value	Adjusted OR (95% CI)	*p*-Value
Nationalities (ASEAN as reference)				
ARAB	12.6 (3.85–41.21)	<0.001	14.68 (3.93–54.91)	<0.001
OTHERS	3.1 (1.56–6.03)	0.001	2.69 (1.26–5.77)	0.011
Sex (Male as reference)				
Female	1.35 (0.75–2.43)	0.3166	0.96 (0.47–1.95)	0.901
Age group (Reference ≤ 15 years)				
>15 but ≤60 years	1.9 (1.04–3.50)	0.037	1.71 (0.81–3.63)	0.161
>60 years	6.4 (0.71–57.14)	0.098	2.98 (0.2–44.59)	0.429
Below-average economic level (45,707 THB) (Above average as reference)				
Below average	3.7 (1.36–9.85)	0.010	3.50 (1.04–11.68)	0.043
Being sick over the past month (No as reference)				
Yes	5.0 (2.57–9.77)	<0.001	4.32 (2.02–9.24)	<0.001
Chronic diseases (No as reference)				
Yes	2.9 (1.32–6.46)	0.008	1.95 (0.73–5.22)	0.183

OR: Odds ratio.

**Table 4 ijerph-18-03901-t004:** Bivariate analysis and multivariate logistic regression for unmet need of in-patient (IP) care.

Factors	Bivariate Analysis		Multivariable Logistic Regression	
	Crude OR (95% CI)	*p*-Value	Crude OR (95% CI)	*p*-Value
Nationalities (ASEAN as reference)				
ARAB	10.8 (3.65–31.94)	<0.001	12.93 (3.92–42.65)	<0.001
OTHERS	2.7 (1.13–6.53)	0.025	2.24 (0.89–5.63)	0.088
Sex (Male as reference)				
Female	1.75 (0.89–3.45)	0.106	1.34 (0.62–2.91)	0.458
Age group (≤15 years as reference)				
>15 but ≤60 years	1.47 (0.74–2.95)	0.46	1.07 (0.46–2.45)	0.877
>60 years	2.2 (0.34–14.24)	0.41	0.45 (0.04–5.51)	0.536
Below-average economic level (45,707 THB) (Above average as reference)				
Below average	1.75 (0.56–5.50)	0.34	0.72 (0.2–2.63)	0.624
Being sick over the past month (No as reference)				
Yes	4.2 (1.76–10.12)	0.001	3.49 (1.34–9.08)	0.011
Chronic diseases (No as reference)				
Yes	2.4 (1.11–5.08)	0.025	2.84 (1.06–7.6)	0.038

## Data Availability

Data available on request due to ethical restrictions.

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
