# Peer review of "A Cross-Sectional Study on Disparities in Unmet Need among Refugees and Asylum Seekers in Thailand in 2019"

_ijerph, 2021, doi:10.3390/ijerph18083901_

Round 1
Reviewer 1 Report
Thank you for the opportunity to review your manuscript. The study addresses an important subject - that of unmet healthcare need in urban refugees and asylum seekers. It builds on work from the same study recently published by the Authors (Suphanchaimat, R., et al. 2020 https://doi.org/10.1186/s12939-020-01316-y). While the findings in themselves are not highly novel, they add to the growing body of evidence that shows that URAS from Arab countries have greater unmet healthcare need and may inform local health policy and interventions. The study was well designed, well analysed and well reported. I have a few minor comments/suggestions:
- The manuscript requires further proofing. There remain numerous grammatical errors, largely related to tense and plurals.
- It was unclear how large the URAS population in Bangkok is, nor whether the survey sample was broadly representative of the population as a whole, and of the subgroups (Arab, ASEAN, others).
- It was also unclear if all the respondents had lived in Bangkok for the 12 months preceding participation in the study.
- I found the demographic data a little confusing, particularly items relating to whether URAS had been sick. This was variably reported as in the past month (table 1), six months (lines 208-209) and differed from the measure of unmet need in the past 12 months. It would be interesting to know the number and proportion of respondents who reported being unwell at each of these time points.
- I think the data reported in figure 1 would be better presented in a table to allow for number of respondents, as well as percentage, to be reported, especially as some of the category sizes are quite small. If you want to keep the figure, I suggest making the shades representing unmet and no unmet need the same across both charts.
- There is a little duplication in the discussion, particularly around the discussion of the Arab group having high unmet needs (e.g. paragraph beginning line 313) that could perhaps be refined.
Author Response
Thank you very much for your constructive comments. Please kindly find my clarification regarding your comments as follows.
The manuscript requires further proofing. There remain numerous grammatical errors, largely related to tense and plurals.
- Thank you for your suggestion. I have done proofreading again to make it more consistent across the whole article with assistance from native speakers.
It was unclear how large the URAS population in Bangkok is, nor whether the survey sample was broadly representative of the population as a whole, and of the subgroups (Arab, ASEAN, others).
- Please kindly find the update of a flowchart on sampling frame. In this study, the samples were drawn from 3,021 from the top-ton nationalities (n = 181) by stratified random sampling.
It was also unclear if all the respondents had lived in Bangkok for the 12 months preceding participation in the study
- The information on years in Thailand means the time that they spent in the BRC. It is quite limited to track the data back due to a privacy concern.
I found the demographic data a little confusing, particularly items relating to whether URAS had been sick. This was variably reported as in the past month (table 1), six months (lines 208-209) and differed from the measure of unmet need in the past 12 months. It would be interesting to know the number and proportion of respondents who reported being unwell at each of these time points
I would like to apologize for this unclear point. This is a typo (as see in LINE 199). Our question is asking if during one month before the interview they got sick or not. So the exact timeframe here is being sick over the past month.
I think the data reported in figure 1 would be better presented in a table to allow for number of respondents, as well as percentage, to be reported, especially as some of the category sizes are quite small. If you want to keep the figure, I suggest making the shades representing unmet and no unmet need the same across both charts.
Thank you. I have changed it to Table 2. Please kindly find it in P6.
There is a little duplication in the discussion, particularly around the discussion of the Arab group having high unmet needs (e.g. paragraph beginning line 313) that could perhaps be refined.
Thank you for your suggestion. I have revised it and please kindly find it in LINE 311-312.
Reviewer 2 Report
Thanks so much for examining a deprived population. It was concise in general. However, you need to check your grammatical mistakes. You may use editing services.
Additionally, you may also cite from studies about URAS in Turkey which consists a big proportion of the entire URAS population.
I did not see a clear framework to discuss your findings. Rather, I think you seem to report your findings in the discussion section. I suggest you take a framework which will help highlight the importance of your findings to analyze.
Author Response
Thank you very much for your feedback. I have done a revision regarding your comments below.
Thanks so much for examining a deprived population. It was concise in general. However, you need to check your grammatical mistakes. You may use editing services.
- Thank you for your comment. I have done proofreading again to improve my grammatical error and consistency.
Additionally, you may also cite from studies about URAS in Turkey which consists a big proportion of the entire URAS population.
- Thank you. I have more information on Turkey and refugees in LINE 53-56.
I did not see a clear framework to discuss your findings. Rather, I think you seem to report your findings in the discussion section. I suggest you take a framework which will help highlight the importance of your findings to analyze.
- Thank you for your critical point. Here, I try to follow the framework of Zimmerman et al (2011) which discussed about migrants’ travel and social determinants of migrants. So in the discussion part I attempted to link the findings to the point of social determinants of health, please see LINE 317-325.
Reviewer 3 Report
The article by Kunpeuk et al, titled "A cross-sectional study on disparities in unmet need among refugees and asylum seekers in Thailand in 2019" investigates the unmet healthcare needs of the refugees who were divided into three categories based on their country of origin. The study indicates a correlation between Arab nationality and incidence of umnet health care needs.
Strengths:
- Well laid out methodology and defined statistical analysis..
- Rationale for the analysis adequately described
Comments:
- Figure legends need to be corrected.
- More information on chronic diseases affecting the refugees and asylum seekers, age distribution and its timeline with respect to migration may be be helpful understanding the challenges faced by refugees to seek medical care.
Author Response
Thank you very much for your suggestion. Now I have revised my draft regarding your comments and please kindly find it as follows.
Figure legends need to be corrected.
- Thank you for your comment. Now I have changed my figure to Table 2 regarding one reviewer’s suggestion. Please kindly find it in Page 6.
More information on chronic diseases affecting the refugees and asylum seekers, age distribution and its timeline with respect to migration may be helpful understanding the challenges faced by refugees to seek medical care.
- Thank you for your comment. I have added more information about mean and median age in Table1. As the information on migration years is quite limited due to a privacy concern, we did not present the data in this article. Besides, the info about years in Thailand is likely to be inaccurate since we have the data about urban refugees and asylum seekers once only after their registration with UNHCR (not from the first day they arrived in Thailand).
Reviewer 4 Report
Thank you for this interesting and relevant work. The paper is well written and needs some clarification and reductions.
Abstract: please give a usual structure to your abstract. E.g. beginn with one sentence for describing the background and the current state of literature.
Introduction
Please add one sentence about the setting in Thailand concerning outpatients and inpatients
Methods
Please describe more in detail the emergence oft he formula: did anyone else used it beforehand? Are the basic assumption of 11% a good basis for calculating, if it is only based on one paper from italy? In which way the health-care systems between Thailand and Italy are different? Please add this probleme (different health systems) also into the discussion section.
2.1 study Design: it is not fully understandable, why less than 10% (206) of the total number (3021) are coming from the top ten countries. What about the 25 drop outs from the first step?
Please add a flowchart which shows the recruitment-procedere.
How was the questionnaire distributet between the participants? Paper/pencil? Phone? Internet? Interview? How much from which method?
Results
Please shorten the first paragraph oft he „result-section“ and refer to the tables.
Please shorten also the section 3.3 and refer to table 2.
The 1st column in all tables should be left-aligned
If possible add more subgroups of ages. In refugees population we mostly see younger persons, the classification into only 3 groups is inaccurate. Please add a group 15-25 years, 25-35 and above 35 up to 60. If not possible, please add this into the discussion/limitations.
Discussion
Line 263: please delete „moste were female“ (52%?)
Please add tot he limitation section, that no diagnoses were collected and consider to shorten also the introduction relating to diagnoses.
Author Response
Thank you very much for your constructive comments. I have revised this draft and try to improve the quality of this paper regarding your suggestions. Please kindly find my clarification point-by-point as follows.
Abstract: please give a usual structure to your abstract. E.g. begin with one sentence for describing the background and the current state of literature.
- Thank you for your suggestion. Please kindly find more information on the introduction of this study in Page 1. Health of urban refugees and asylum seekers (URAS) in Thailand has been under-researched compared with other groups of non-Thai populations, especially in the topic of unmet need.
Introduction, please add one sentence about the setting in Thailand concerning outpatients and inpatients.
- Thank you for this point. I have added information about in-patient and out-patient care in Thai population in the discussion as I think I could get along well with the results of OP and IP care of URAS, please see LINE 277 -280.
Please describe more in detail the emergence of the formula: did anyone else used it beforehand? Are the basic assumption of 11% a good basis for calculating, if it is only based on one paper from Italy? In which way the health-care systems between Thailand and Italy are different? Please add this problem (different health systems) also into the discussion section.
- We used the 11% figure from the Italian study as that study used the same unmet need question like our study. We tried to explore other similar studies in Asia, but unfortunately, we found none. This also reflected the dire need of research on this population in Asia. The main healthcare system in Italy is financed by tax like Thailand. However, most health facilities in Italy are governed by the local governments. We have discussed this point in LINE 328-342.
study Design: it is not fully understandable, why less than 10% (206) of the total number (3021) are coming from the top ten countries. What about the 25 drop outs from the first step?
- Thank you for your comments, 25 drop outs were from refusing to participate and incomplete response. I have added the flowchart of recruitment process, please see the supplementary file.
Please add a flowchart which shows the recruitment-procedure.
- Thank you. I have added a recruitment process in the supplementary file.
How was the questionnaire distributed between the participants? Paper/pencil? Phone? Internet? Interview? How much from which method?
- The questionnaires were distributed by BRC interpreter volunteers with paper-based interview. However, the participants who were not willing to come to the BRC due to their difficulties were interviewed via phone interview. In total, 159 participants (87.8%) completed the survey by face-to-face interview whereas 22 participants (12.2%) were interviewed by phone.
Please shorten the first paragraph of the result-section and refer to the tables.
- Thank you and please kindly find it in LINE 194.
Please shorten also the section 3.3 and refer to table 2.
- Please find it in LINE 230-238.
The 1st column in all tables should be left-aligned
- I have improved info in the table according to this format.
If possible add more subgroups of ages. In refugees population we mostly see younger persons, the classification into only 3 groups is inaccurate. Please add a group 15-25 years, 25-35 and above 35 up to 60. If not possible, please add this into the discussion/limitations.
- Thank you for your comment. The subgroup we did aimed to categorized participants into the younger-age group, the working-age group, and the elderly. These groups will face different degree of unmet need in the Thai context and we hope it could inform some policy messages in the country.
Line 263: please delete, most were female“ (52%?)
- Thank you for correcting the typo. We will do proofreading again to improve spelling and grammatical error.
Please add the limitation section, that no diagnoses were collected and consider to shorten also the introduction relating to diagnoses.
- Thank you for your comment. I have added limitation on diagnosis as you suggested in the discussion part in LINE 350.